# Combined Gastric and Colorectal Cancer Screening—A New Strategy

**DOI:** 10.3390/ijms19123854

**Published:** 2018-12-03

**Authors:** Michael Selgrad, Jan Bornschein, Arne Kandulski, Jochen Weigt, Albert Roessner, Thomas Wex, Peter Malfertheiner

**Affiliations:** 1Department of Gastroenterology, Hepatology and Infectious Diseases, Otto-von-Guericke-University of Magdeburg, Leipziger Str. 44, 39120 Magdeburg, Germany; michael.selgrad@ukr.de (M.S.); jan.bornschein@ouh.nhs.uk (J.B.); arne.kandulski@ukr.de (A.K.); jochen.weigt@med.ovgu.de (J.W.); t.wex@schenk-ansorge.de (T.W.); 2Department of Internal Medicine I, University Hospital of Regensburg, Franz-Josef-Strauß-Allee 11, 93053 Regensburg, Germany; 3Translational Gastroenterology Unit, John Radcliffe Hospital, Oxford University Hospitals, Headley Way, Oxford OX3 9DU, UK; 4Department of Pathology, Otto-von-Guericke-University of Magdeburg, Leipziger Str. 44, 39120 Magdeburg, Germany; albert.roessner@med.ovgu.de; 5Medical Laboratory for Clinical Chemistry, Microbiology and Infectious Diseases, Department of Molecular Genetics, Schwiesaustr. 12, 39124 Magdeburg, Germany

**Keywords:** gastric cancer, *Helicobacter pylori*, screening, pepsinogen, screening colonoscopy

## Abstract

Background: Our aim was to evaluate the feasibility of a serological assessment of gastric cancer risk in patients undergoing colonoscopy in countries with low-to-moderate incidence rates. Methods: Serum samples were prospectively collected from 453 patients (>50 years old) undergoing colonoscopies. Of these, 279 (61.6%) also underwent gastroscopy to correlate the results for serum pepsinogen I and II (sPG-I and sPG-II), sPG-I/II ratio, and anti-*H. pylori* antibodies with gastric histopathology findings (graded according to the updated Sydney classification and the Operative Link of Gastritis Assessment (OLGA) and the Operative Link for Gastric Intestinal Metaplasia assessment (OLGIM) systems). Results: *H. pylori* was found in 85 patients (30.5%). Chronic atrophic gastritis was diagnosed in 89 (31.9%) patients. High-risk OLGA (III–IV) stages were present in 24 patients, and high-risk OLGIM stages were present in 14 patients. There was an inverse correlation of sPG-I with the degree of atrophy and intestinal metaplasia (IM), as well as with the respective OLGA (r = −0.425; *p* < 0.001) and OLGIM (r = −0.303; *p* < 0.001) stages. A pathological sPG-I result was associated with a relative risk (RR) of 12.2 (95% confidence interval: 6.29–23.54; *p* < 0.001) for gastric preneoplastic changes. Conclusions: The assessment of serum pepsinogen allows the identification of patients at increased risk of gastric cancer. A prevention strategy of combining a screening colonoscopy with a serological screening for preneoplastic gastric changes should be considered in the general population.

## 1. Introduction

Screening programs for colorectal cancer are well established and based on varying approaches for implementation in many parts of the world [1]. Screening colonoscopies have led to a significant reduction in the incidence and mortality of colorectal cancer [2,3,4]. Even in countries with low acceptance rates, screening has been shown to prevent colorectal cancer by 31–64% [5]. In contrast to the established and effective prevention strategies for colorectal cancer, a strategy for gastric cancer prevention has been introduced in only a few countries, Japan and Korea, both of which have high incidences of gastric cancer. Prevention strategies for gastric cancer were initially based on radiography with encouraging results. Today, screening in these countries is performed by endoscopy and serology and a combination of both [6,7].

There are two main reasons to consider gastric cancer screening in low-to-moderate incidence areas: First, gastric cancer remains one of the leading causes of cancer-related deaths, with more than 900,000 cases per year [8,9]. Second, the strong association between *Helicobacter pylori* (*H. pylori*) and gastric cancer offers the unique chance for the primary prevention of gastric carcinogenesis by eradication of *H. pylori*, because 89% of all gastric neoplasia cases are attributable to this infection [10,11,12,13,14]. *H. pylori*-induced gastric carcinogenesis is a multi-step process, which evolves in many cases from chronic gastritis to preneoplastic changes (atrophic gastritis and intestinal metaplasia) and ultimately into gastric cancer [15]. Therefore, the goal of gastric cancer prevention should be the earliest possible elimination of the infection by eradication therapy, ideally prior to the development of preneoplastic changes [16]. Patients in whom preneoplastic changes of the gastric mucosa are identified should be subject to meaningful surveillance programs, comparable to polyp surveillance for colorectal cancer prevention [17]. A subset of patients with preneoplastic changes will also benefit from eradication therapy, which has the potential to interrupt the progression towards neoplastic stages [18]. A simple blood test with the determination of serum pepsinogen I and II (sPG-I and sPG-II), including the calculation of the sPG-I/II ratio, in combination with the analysis of anti-*H. pylori* antibodies, allows for the identification of patients with severe atrophic gastritis either with or without *H. pylori* infection, who are at an increased risk for gastric cancer [19,20]. These high-risk patients should then be invited for a diagnostic gastroscopy to assess the presence of preneoplastic (or neoplastic) changes of the gastric mucosa and to determine if they are candidates for further surveillance [20,21]. While a population-based gastric cancer screening is highly effective in countries with high incidences of this tumour, it is unlikely to be cost-effective in countries with a lower gastric cancer rate [22,23,24,25,26]. For these countries, we propose the approach of a serological “pre-screening” that could be combined with an already established colorectal cancer screening program.

In the current study, we analysed the feasibility of using serological markers for the identification of patients with preneoplastic changes of the gastric mucosa among those who were undergoing colonoscopies.

## 2. Results

### 2.1. General Characteristics of the Study Population

Of the 453 individuals enrolled in this study, 196 (43.3%) were women and 257 (56.7%) were men with a mean age of 67.0 years (9.96 years standard deviation). The reasons for the colonoscopies in these individuals were colorectal cancer screening, including surveillance following a polypectomy (*n* = 187), and diagnostic indications such as abdominal pain (*n* = 47), altered bowels (*n* = 51), and per-anal bleeding (*n* = 40). The remaining 128 patients were referred for various other reasons. Of all 453 individuals, 279 also received a diagnostic upper gastrointestinal endoscopy (61.6%) and were therefore included in the ‘core cohort’ for further analysis. The distribution of sex, age, and reason for colonoscopy were not different between the total study population and the core cohort.

### 2.2. Histopathological Results of Endoscopic Gastric Biopsies

Of the 279 patients who received an upper gastrointestinal endoscopy, 277 had complete biopsy sets for the characterisation of gastric inflammation and atrophic changes according to the updated Sydney classification. Preneoplastic changes were diagnosed in 89 (32.1%) of the patients. In 64 patients (23.1%), atrophic changes of the gastric mucosa were seen, with a comparable distribution between the antrum and the body. Of these, 24 patients were classified as high-risk Operative Link of Gastritis Assessment (OLGA) stages III and IV (8.7%). In 59 patients (21.3%), IM was present, more often focused in the antrum than in the body (17.0% versus 8.3%). High-risk Operative Link for Gastric Intestinal Metaplasia assessment (OLGIM) stages were seen in 14 (5.1%) patients. In 35 patients (12.6%), active gastric inflammation was seen without preneoplastic changes.

### 2.3. Helicobacter Pylori Infection Status

A total of 85 patients (30.5%) in the core group was *H. pylori*-positive by serology (32.9% of the total study population). *H. pylori*-positive patients showed significantly more active gastric inflammation (54.7% versus 21.0% *p* < 0.001) and more preneoplastic changes compared with patients without the infection (41.2% versus 27.8%; *p* = 0.036). Although *H. pylori* status was not statistically associated with the presence of gastric atrophy (*p* = 0.437), there was a higher rate of patients with OLGA high-risk stages among the *H. pylori*-positive patients (15.5% versus 5.6%; *p* = 0.01). This could not be confirmed for the OLGIM stratification (*p* = 0.756).

### 2.4. Serum Pepsinogen I and Histopathological Alterations

There was the expected inverse correlation of serum levels of sPG-I and the degree of atrophy and IM in the gastric antrum and body (*p* < 0.001). This correlation was confirmed for the respective stages according to the OLGA (r = −0.494; *p* < 0.001) and OLGIM (r = −0.325; *p* < 0.001) system (Figure 1). There was furthermore a weak correlation of sPG-I with the age of the patients (r = −0.145; *p* = 0.016).

As mentioned above, a cut-off of sPG-I below 50 ng/µL was considered to be pathological in accordance with the current literature. A total of 60 patients presented sPG-I levels below this threshold. Of these, 45 (75%) showed preneoplastic changes on histology, compared with 44 of the 219 sPG-I-negative patients (20.1%; *p* < 0.001). There were more sPG-I-positive patients among those with active inflammation in the gastric body (35.6% versus 19.0%; *p* = 0.013), which could not be confirmed for inflammation in the antrum (*p* = 0.168). Of the 60 sPG-I-positive subjects, 42 (70.0%) showed some degree of glandular atrophy (*p* < 0.001) independent from the location, and 28 (46.7%) were positive for IM (*p* < 0.001). These results are summarised in Table 1.

There was no association between serum pepsinogens and the reasons for colonoscopy in our study population.

### 2.5. Combination with Other Serum Parameters

The diagnostic performance of the assessment of the sPG-I/II-ratio was significantly inferior compared with sPG-I alone with regards to the detection of gastric preneoplastic changes. The analysis of patients that were positive in both approaches (as “true positives”) improved the diagnostic quality of the blood test. Sensitivity, specificity, the positive predictive value, and negative predictive value for gastric atrophy changed from 65.6%, 91.5%, 70.0%, and 89.9% for sPG-I alone to 68.8%, 91.9%, 69.8%, and 90.7%, for the combined assessment, respectively.

The ABCD screening method, which was developed for risk assessment for gastric cancer development, includes the serological *H. pylori* status in the overall risk stratification [20,27]. In our cohort, the inclusion of the serological *H pylori* status had no effect on the results.

### 2.6. Risk Stratification

A positive (i.e., abnormally low) serum pepsinogen test was associated with a relative risk (RR) of 12.2 (95% confidence interval (CI): 6.29–23.54) for gastric atrophic changes. It must be noted that in patients with a positive serum test, the RR for glandular atrophy was higher (22.5, 95% CI: 11.07–45.60) than the risk for IM (5.8, 96% CI: 3.09–10.90). As mentioned above, the risk of progression towards gastric cancer that is associated with the degree and location of both atrophy and IM can be stratified according to the OLGA or the OLGIM system. The RR for high-risk OLGA stages III and IV in patients with a positive pepsinogen test was 13.9 (95% CI: 5.21–36.89; *p* < 0.001), and 14.9 (95% CI: 4.01–55.26; *p* < 0.001) for the respective OLGIM stages.

Patients with high-risk OLGA stages could be identified by serum pepsinogen assessment with a sensitivity of 75.0%, a specificity of 82.2%, a positive predictive value (PPV) of 28.6%, and a negative predictive value (NPV) of 97.2%; high-risk OLGIM patients were at 78.6%, 80.2%, 17.5%m and 98.6%, respectively.

### 2.7. Proposal

Figure 2 outlines a proposed algorithm and demonstrates a possible reflection of our results in a population-based screening program. In our study population, 19.4% had a pathological pepsinogen test (i.e., *n* = 194 of 1000 patients who are screened when presenting for a colorectal cancer screening program). If these patients are then invited and undergo gastroscopy and biopsy sampling, 135 patients would be expected to have gastric atrophy, and 55 of these would be in high-risk stages according to the OLGA staging (the numbers for IM and OLGIM are not calculated for this example). Only these 55 patients (5.5% of the original population) would be invited into an endoscopic surveillance regimen of three yearly endoscopies, as suggested by the European guidelines [17]. All the other patients would be advised to have a repeat blood test after a 10-year interval (the interval could be shortened if cost-effective). By this approach, 23 of every 1000 subjects who would benefit from surveillance would be missed, as opposed to the total of 78 patients who would be missed without any screening program.

The cost-effectiveness of this approach depends on the local (national) availability and costs of an upper gastrointestinal endoscopy and the implemented screening and prevention program for colorectal cancer.

## 3. Discussion

The application of a serum test to screen for significant atrophic (preneoplastic) changes of the stomach in patients undergoing colonoscopies identified 19% of the screened individuals as being at risk for gastric cancer. The serum test delivers two important results that are crucial for further risk stratification: (a) information on the likelihood of the presence of preneoplastic gastric mucosal changes, and (b) information on one’s *H. pylori* infection status. The combination of subsequent *H. pylori* eradication, as well as endoscopic confirmation of preneoplastic changes and further endoscopic surveillance, allow for effective primary and secondary prevention of gastric cancer. A combined approach with a screening colonoscopy would allow for an even more structured strategy. It is of note that 24% of our patients had low-grade adenomas, whereas 3.7% showed high-grade lesions and adenocarcinomas of the colon.

In our study, almost a third of the patients showed altered pepsinogen values in the serum, and in 75% of those who underwent OGD, preneoplastic changes were confirmed by histology. These numbers were higher compared with a similar study that has recently been published on a Slovenian population [28]. Tepes et al. applied, however, a stricter cut-off (<30 ng/mL) for abnormal sPG-I values and selected, therefore, fewer patients for endoscopies. The risk of such an approach is to lose sensitivity by gaining specificity, but missing a patient at risk or with early gastric cancer seems to overcompensate for an additional number of patients who undergo OGD without high-risk gastric pathology. In the study by Tepes et al., the prevalence of *H. pylori* by serology was much higher compared with our cohort, so stricter cut-offs might have been necessary to avoid referring the majority of the general population in the respective region for endoscopies [28]. However, patients with *H. pylori* antibodies at a higher age are in any case at increased risk. An endoscopy will help to select those who require further surveillance. The overall performance of the approach was similar to our study, although the incidence of gastric cancer is lower in Germany. It must be mentioned that we also assessed serum levels of Gastrin-17 in our cohort, but the results did not show any association with clinical or demographic parameters [29,30,31]. There could be bias, because Gatsrin-17 is an unstable polypeptide that requires highly standardised conditions for sampling and processing. Therefore, we would not consider Gastrin-17 as a factor that should be included in a routine screening approach. Interestingly, De Re et al. showed a good correlation of Gastrin-17 values with OLGIM status, therefore considering this parameter as a good risk predictor for gastric preneoplastic lesions [32]. It must be noted, however, that this study analysed data from a very specific cohort with a primary target group of patients with auto-immune chronic atrophic gastritis, and further assessment is required to determine if these results are also applicable to patients with changes that are not related to auto-immune disease. The study by De Re and colleagues might have also had the advantage of having obtained the blood samples under strictly standardised conditions, as such conditions are necessary for the assessment of Gastrin-17. This was not feasible for our cohort of patients, who were booked for routine colonoscopies, although we tried to implement a standardised approach.

Our study was not designed to provide evidence that our strategy prevents gastric cancer in a low-incidence setting but was rather as a feasibility approach. Therefore, we also did not undertake precise Receiver-Operating-Characteristic (ROC) analyses to determine the ideal cut-off threshold for sPG-I in view of its diagnostic purposes as done by others [32]. The cut-off values often reflect regional differences in the respective patient population, and local optimisation should be considered when this strategy is applied in clinical practice. In addition, long-term follow-up of a larger population of patients with preneoplastic changes is needed and is underway in several European countries. Therefore, the identification of patients with preneoplastic changes of the stomach by targeted endoscopic investigation not only excludes malignancy but, more importantly, allows these patients to enter a clinical, meaningful surveillance pathway, aimed at the early detection of gastric cancer, as proposed by several guidelines [17,24]. This is important for countries with a low incidence of gastric cancer, in which this disease is mostly diagnosed at an advanced stage of the disease, when there are more limited treatment options and a five-year survival rate ranging from 15% to 30% [33].

Another important aspect of our study is the identification of patients with the *H. pylori* infection, which affects approximately one-third of the patients. This offers the chance for the primary prevention of gastric cancer by offering those patients eradication therapy. One may argue that our patient cohort is probably too old for a primary prevention approach, but *H. pylori* eradication can still have an effect even at a stage when either preneoplastic or even early neoplastic changes have developed [34,35,36,37]. While the data on the regression and improvement of preneoplastic changes of the stomach after eradication therapy are conflicting [13,38,39,40,41], *H. pylori* eradication after endoscopic resection of early gastric cancer has been effective in preventing the development of metachronous gastric carcinoma in a subset of patients [42,43]. Therefore, eradication therapy should be offered to patients with preneoplastic changes of the stomach, combined with regular endoscopic surveillance as follow-up [35]. The combination a serological “pre-screening” with a colonoscopy could improve the cost-effectiveness of such an approach, even in populations with low or intermediate gastric cancer risk [44].

A group from Porto recently calculated the cost-effectiveness of combining screening gastroscopy with colorectal cancer screening. While a single-approach gastroscopy was clearly not cost-effective in the general population, the combination with a screening colonoscopy as listed in our proposal was cost-effective; however, this was only true when the regional gastric cancer incidence was higher than 10/100,000 per year [26]. To use the serum test as a pre-screening tool to lower the numbers of those requiring an endoscopic assessment could make such a strategy work in countries with even lower incidences, such as Germany. In studies from Asia, the combination of a serum assessment of pepsinogen I and *H. pylori* have been reported to be cost-effective for identifying people at risk for gastric cancer [45,46]. The pepsinogen test method, which is widely employed in Asian countries, has also been proven to be effective in western countries, with similar accuracy estimates as compared with Asia, underlining its applicability as an effective screening tool in Europe as well [47].

It must be noted that the combination of screening procedures for both gastric and colorectal cancer was well accepted by the patients in our study.

In conclusion, our study demonstrates that the opportunity to combine colon cancer prevention with the serological assessment of pepsinogen-I and *H. pylori* status can easily be achieved and allows for the identification of patients with advanced gastric atrophic changes at increased risk for gastric cancer. Future studies with a long-term follow-up are needed to validate our results in larger cohorts, ideally as a multinational approach to analyse the impact of different health care settings.

## 4. Material and Methods

### 4.1. Study Cohort

A total of 453 individuals with a minimum age of 50 years was prospectively enrolled in this study. These individuals were referred for colonoscopies at the Department of Gastroenterology, Hepatology, and Infectious Diseases of the University of Magdeburg, Germany. The local Ethics committee of the Otto-von-Guericke University, Magdeburg, Germany approved the study (63/08; 13 November 2008). The study was conducted according to the Declaration of Helsinki (as revised in 1989). All the patients gave their written informed consent.

The primary analysis was undertaken on a core group of *n* = 279 patients, who also underwent an upper gastrointestinal endoscopy for various indications. In this group, the performance of the serological test was evaluated with regards to the endoscopic and histopathological findings in the stomach. Based on these data, the feasibility of a broader approach was estimated for the complete study cohort.

### 4.2. Assessment of Serum Parameters

Prior to the endoscopic procedure, a blood sample (5–7 mL) was collected from each patient during a fasting state. Serum that was prepared by centrifugation at 7000× *g* at 4 °C for 15 min was aliquoted in three individual cryotubes (each 1–1.5 mL) and then stored at −80 °C for further analysis. An *H. pylori* IgG enzyme-linked immunosorbent assay (Biohit, Rosbach, Germany) was used for the analysis of the *H. pylori* status according to the manufacturer’s instructions. The patients were classified as *H. pylori*-positive based on the presence of the *H. pylori*-specific IgG (≥30 enzyme immunounits (EIU)), whereas values below this threshold led to the assignment of an *H. pylori*-negative status. The analyses of sPG-I, sPG-II (Pepsinogen-I, Pepsinogen-II EIA Test Kit, Biohit Plc, Helsinki, Finland) was performed on the same aliquot as the assessment of the *H. pylori* status according to the manufacturer’s instructions and as described previously [29]. A sPG-I level of <50 ng/mL and a sPG-I/II ratio of <3 were considered as pathological results and indicative of advanced atrophy of the gastric mucosa.

### 4.3. Histopathological Assessment

Patients with a pathological serum test result were invited for a diagnostic gastroduodenoscopy. As indicated above, results from patients who underwent gastroscopy for other indications have also been included in this analysis. Gastric biopsies were processed by routine methods. One section of each sample was stained with haematoxylin and eosin, modified Giemsa for the detection of *H. pylori*, and Periodic acid-Schiff (PAS) stain. A histological assessments of the gastric biopsies was performed by expert upper gastrointestinal pathologists. Histopathological alterations of the gastric mucosa were assessed according to the updated Sydney system [48]. The extent and degree of atrophic gastritis and intestinal metaplasia (IM) in samples from the gastric antrum and body were used to stratify patients according to the OLGA and OLGIM staging classifications, respectively [49,50].

### 4.4. Statistical Analysis

All statistical analyses were performed using SPSS19.0 for Windows (IBM SPSS Statistics, IBM Corp., Armonk, NY, USA). Student´s *t*-test was used for the group-wise comparison of the parametrical data (age), and the Mann–Whitney U test was used for non-parametrical comparisons (serological parameters). The categorical data were compared by Fisher’s exact test, also applying Pearson’s χ^2^ for risk assessment. For all the tests, a two-sided significance level of *p* < 0.05 was considered as statistically significant.

## Figures and Tables

**Figure 1 ijms-19-03854-f001:**
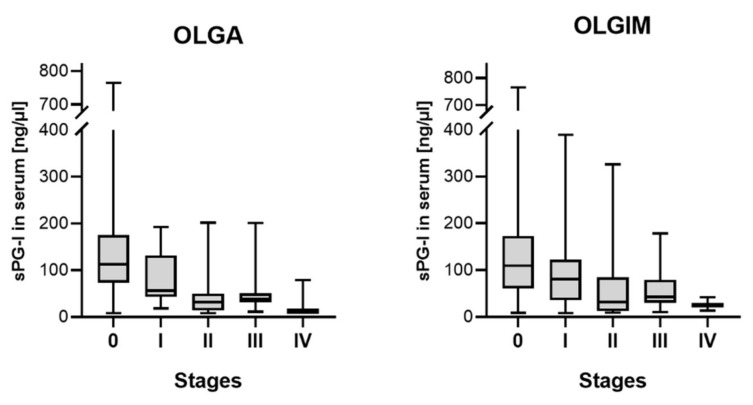
Serum concentration of pepsinogen I according to the respective Operative Link of Gastritis Assessment (OLGA) (*p* < 0.001) and Operative Link for Gastric Intestinal Metaplasia Assessment (OLGIM) (*p* < 0.001) stages.

**Figure 2 ijms-19-03854-f002:**
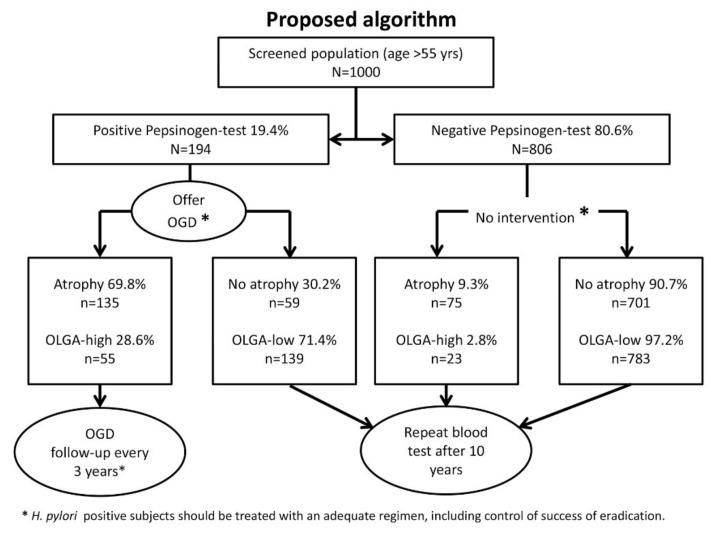
Proposed algorithm for a population-based screening program. Outlined are the assumptions for a population of *n* = 1000 individuals based on the results of our study. See the main text for further details.

**Table 1 ijms-19-03854-t001:** Analysis of pepsinogen status in the serum and histopathological alterations.

Item		PG*-Positive (*n* = 63)	PG-Negative (*n* = 216)	Total (*n* = 279)	*p*-Value
**Age**	**median (range)**	64 (50–93)	67 (50–94)	67 (50–94)	0.931
**Sex**	**male (%)**	37 (58.7%)	121 (56.0%)	158 (56.6%)	0.773
***H. pylori***	**positive (%)**	21 (33.3%)	64 (29.6%)	85 (30.5%)	0.641
**Atrophy**	**positive (%)**	44 (69.8%)	20 (9.3%)	64 (23.1%)	<0.001
**IM ***	**positive (%)**	30 (47.6%)	29 (13.6%)	59 (21.3%)	<0.001
**OLGA ***	**Stage 0**	17 (27.0%)	194 (90.7%)	211 (76.2%)	<0.001
	**Stage I**	11 (17.5%)	9 (4.2%)	20 (7.2%)	
	**Stage II**	17 (27.0%)	5 (2.3%)	22 (7.9%)	
	**Stage III**	16 (25.4%)	5 (2.3%)	21 (7.6%)	
	**Stage IV**	2 (3.2%)	1 (0.5%)	3 (1.1%)	
**OLGIM ***	**Stage 0**	33 (27.0%)	185 (86.4%)	218 (78.7%)	<0.001
	**Stage I**	9 (17.5%)	19 (6.9%)	28 (10.1%)	
	**Stage II**	10 (27.0%)	7 (3.3%)	17 (6.1%)	
	**Stage III**	8 (25.4%)	3 (1.4%)	11 (4.0%)	
	**Stage IV**	3 (3.2%)	0 (0.0%)	3 (1.1%)	

PG-positive were patients with either PG1 < 50 ng/mL or a PG-ratio < 3.0. * Abbreviations: PG = pepsinogen; IM = intestinal metaplasia; OLGA = Operative Link of Gastritis Assessment; OLGIM = Operative Link for Gastric Intestinal Metaplasia Assessment.

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
