# Peer review of "Combined Gastric and Colorectal Cancer Screening—A New Strategy"

_ijms, 2018, doi:10.3390/ijms19123854_

Reviewer 1 Report

The study is in line with the scientific community. aAuthors repropose the use of a non-invasive appoach for the identification of patients at risk of gastric cancer. The novelty consists in the recruitment of the population in surveillance for colon cancer, and in the reported frequency of presence of atrophy and H. pylori infection in a geographical population with a low incidence of gastric cancer, in the number of samples, in the detailed clinical-histopoltogical examination of injuries and in the multidisciplinary approach. However the cut-off of sPG1<50 and sPG1/sPG2 < 3  could be revaluated based on the gastrin value as proposed in another study (De Re  et al .Clin Trans Gastroenterol 2016 Jul; 7(7): e183) This may have implication int he present study and require at least a comment or better a table with exposed result based on this differnt approach for a comparison of the results 

Author Response

Dear Reviewer,

thank you very much for the constructive comments regarding our paper. We read with interest the publication by De Re et al. and included this paper in the references of our manuscript. Regarding the specific points raised by the reviewer we added the following paragraphs in the discussion section:

-- line 224 and following: 

Interestingly, De Re et al.  show good correlation of Gastrin-17 values with OLGIM status therefore considering this parameter as good risk predictor for gastric preneoplastic lesions. It has to be noted, however, that this study analysed data from a very specific cohort with a primary target group of patients with auto-immune chronic atrophic gastritis and it requires further assessment if these results are also applicable to patients with changes that are not related to auto-immune disease.The study by De re and colleagues might also carry theadvantage of having obtained the blood samples under strictly standardized conditions as it is necessary for assessment of Gastrin-17. This was not feasible for our cohort of patients who were booked for routine colonoscopyalthough we tried to implement a standardized approach.

-- line 234 and following:

Our study was not designed to provide evidence that our strategy does prevent gastric cancer in a low incidence setting but rather as a feasibility approach. Therefore we also did not undertake precise ROC analyses to determine the ideal cut-offthresholdfor sPG-I in view of its diagnostic purposes as done by others[36]. The cut-off values often reflect regional differences in the respective patient population and localoptimizationshould be consideredwhen this strategy is applied in clinical practice.

We would like to confirm that both OLGA (p<0.001) and OLGIM (p<0.001) class (0-IV) correlated positively with sPG-I values, whereas there was no correlation in our study with Gastrin-17 values (p=0.785 and p=0.311, respectively). We are happy to add boxplots on PG1 distribution across OLGA7OLGIM states similar to the ones in the paper by De Re et al if requested. Since the actual diagnostic quality of the assay was not the focus of our study we did not add these in the current draft to keep the manuscript concise. 

As indicated in the modified paragraphs above, we agree with the reviewer that, ideally, regional cut-off should be determined before application in a population in routine clinical practice. 

Thank you once again for these constructive comments. 

With best regards, 

Jan Bornschein

(on behalf of Dr Michael Selgrad and Prof. Dr Dr Peter Malfertheiner)

Reviewer 2 Report

Dear Authors,

According to me, the manuscript ID: ijms-390895 entitled “Combined Gastric and Colorectal Cancer Screening – A New Strategy” by Michael Selgrad, Jan Bornschein, Arne Kandulski, Jochen Weigt, Albert Roessner, Thomas Wex and Peter Malfertheiner is an interesting paper, because colorectal and stomach cancer are ranked within global top cancers. Therefore, colorectal cancer screening and stomach cancer screening are introduced in several countries.

In this article, Authors analysed the feasibility of using serological markers for the identification of patients with preneoplastic changes of the gastric mucosa among those who are undergoing colonoscopy. These data and analyses are very important for the readership. Moreover, manuscript is well structured, the methodology is sound and the results are clearly presented.

I have some suggestions in order to improve paper, which are the following:

1)   Please correct reference numbers in the whole paper according to instructions for authors. In the text, these numbers should be placed in square brackets [ ], and placed before the punctuation; for example [1], [1–3] or [1,3].

2) Introduction: L. 59-65: There are different fonts in the text.

3)   Results: L. 146: 50ng/μl – 50 ng/μl (Please add a space)

Best regards,

Author Response

Dear Reviewers, 

Thank you very much for these positive and encouraging comments. We adjusted the manuscript according to your suggestions as follows and hope that these changes are to your satisfaction.

- Re point 1): Format of the references has been adjusted.

- Re point 2): This has been corrected. 

- Re point 3): This has been corrected.

With best regards, 

Jan Bornschein

(on behalf of Dr. Michael Selgrad and Prof. Dr. Dr. Peter Malfertheiner)

Round  2

Reviewer 1 Report

Authors answered the comments correctly and I think better to add boxplots on PG1 distribution across OLGA7OLGIM states as suggested by the authors themselves
Author Response

Dear colleague, 

the respective figures have been added. We hope that the changes are satisfactory. 

Best regards, 

Jan Bornschein

(on behalf of Dr Michael Selgrad and Prof. Dr. Dr. Peter Malfertheiner)